**Data Availability Statement:** The study used qualitative and quantitative data. Quantitative data: We included a data set as Supporting Information

# Evaluation of an online suicide prevention program to improve suicide literacy and to reduce suicide stigma: A mixed methods study

**Mareike Dreier** [1]*, **Julia Ludwig**[2], **Martin Härter**[1], **Olaf von dem Knesebeck**[2], **Farhad Rezvani**[1], **Johanna Baumgardt**[3], **Nadine Janis Pohontsch**[4], **Thomas Bock**[3], **Sarah Liebherz**[1]

1 Department of Medical Psychology, Center for Psychosocial Medicine, University Medical Center Hamburg-Eppendorf, Hamburg, Germany, 2 Institute of Medical Sociology, Center for Psychosocial Medicine, University Medical Center Hamburg-Eppendorf, Hamburg, Germany, 3 Department of Psychiatry and Psychotherapy, Center for Psychosocial Medicine, University Medical Center Hamburg-Eppendorf, Hamburg, Germany, 4 Department of General Practice and Primary Care, Center for Psychosocial Medicine, University Medical Center Hamburg-Eppendorf, Hamburg, Germany

* m.dreier@uke.de

## Abstract

Low-threshold e-health approaches in prevention to reduce suicide stigma are scarce. We developed an online program containing video reports on lived experience of suicide and evidence-based information on suicidality. We evaluated the program by a mixed methods design. We examined pre-post-changes of program completers (n = 268) in suicide literacy, suicide stigma (self and perceived), and self-efficacy expectation of being able to seek support in psychologically difficult situations using linear mixed models. To examine reported changes and helpful program elements 12–26 weeks after program completion, we content analyzed transcripts of telephone interviews (n = 16). Program completers showed more suicide literacy (Cohen's d = .74; *p* < .001), higher self-efficacy expectations to seek support (d = .09; *p* < .01), lower self-stigma (subscales glorification/normalization: d = -.13, *p* = .04; isolation/depression: d = -.14; *p* = .04; stigma: d = -.10; *p* = .07; n = 168) compared to baseline. We found no significant differences in perceived suicide stigma. We identified lived experience reports, the possibility of sharing own narrative on stigma and suicidality, and information on support as helpful elements. The current online program can increase suicide literacy and self-efficacy expectations to seek support and reduce self-stigma. We recommend a larger randomized controlled trial with longer follow-up to confirm these findings.

## Introduction

Worldwide, 703 000 people die by suicide every year [1]. Effective treatment options for underlying mental disorders and specifically for suicidality exist [2]. Help seeking can prevent aggravation of psychological problems and even be vital to the survival of a suicidal person [3].

(S1 Data and S2 Data). Qualitative data: Since the data is sensitive (linking of suicidality, other health-related and personal data), there are ethical restrictions on sharing a de-identified data set, i.e., interview transcripts. The participants in the study were informed that the data may only be analyzed with the cooperation of the University Medical Centre Hamburg-Eppendorf. The participants agreed that excerpts of the interviews could be published, which means that an entire interview could not be published. With the corresponding informed consent, the ethics vote was also obtained from the Ethics Committee of the Hamburg Medical Cham-ber (website: https://www.aerztekammer-hamburg.org/ethik_kommission.html; email: ethik@aekhh.de) and a positive decision was made (process number: PV5750).

**Funding:** The German Federal Ministry of Health (in German: Bundesministerium für Gesundheit) funded this study (ZMVI1-2517FSB117, funding period: 10/2017 to 12/2020, funder website: https://www.bundesgesundheitsministerium.de/en/index.html). SL, MH, OvK, TB applied for the grant. The funders had no role in study design, data collection and analysis, decision to publish, or preparation of the manuscript.

**Competing interests:** MD, JL, NP, FR, MH, OvK, JB, TB, and SL declare that they have no competing interests.

Nevertheless, many people affected by suicidality do not seek help [4]. Insufficient information on mental conditions, on support offers, and negative attitudes towards help-seeking, stigma associated with suicidality, i.e., internalized, anticipated, experienced or perceived suicide stigma may decrease help-seeking [5–12].

Suicide stigma and the attached taboo are obstacles in suicide prevention. Raising awareness is important to make progress in preventing suicides [13, 14]. In general, public stigma can be reduced by education [15] and contact between members of a stigmatizing and stigmatized group [16]. Self-stigma in persons with a mental condition can be reduced by education and psychoeducation, therapeutic approaches, empowering, and self-help [17, 18].

Mental health literacy, defined as the "knowledge and beliefs about mental disorders which aid their recognition, management or prevention", enables individuals to seek appropriate help [19, 20]. Rates of help-seeking among people with suicidal ideation are low [21]. Suicidal ideation may not be perceived by the individual as a mental health problem [22]. A lack of knowledge about a mental health condition (e.g., how to recognize or prevent a mental disorder) may also contribute to stigma towards the affected group [10]. Increased literacy may lead to less stigma and more openness to seek treatment for mental disorders [23]. In a German population sample, suicide literacy was negatively associated with stigmatizing attitudes toward suicidal persons, so disseminating knowledge may help to reduce suicide stigma [24].

A person's self-efficacy expectations, i.e., the judgement of one's capability, partly explains the change or maintenance of certain behaviors [25]. In suicide prevention, communicating suicidality and seeking support can be an important behavior for individuals in a suicidal crisis and can ultimately be life-saving [2, 14]. Specific aspects of self-efficacy expectancy, such as confidence in being able to confide in others and being able to seek support in psychologically difficult situations, could be modifiable through a brief program [26, 27].

Online interventions are a low-threshold opportunity to reach many persons who cannot access care in traditional ways. A meta-analysis on internet-based programs which included 16 studies, showed that online interventions can help to reduce suicidal ideation [28]. A systematic literature review found only seven published studies on online self-help interventions that aimed at reducing self-stigma in persons with mental health problems; one targeting suicide attempt related-personal stigma [18]. Although mental illness stigma and suicide stigma overlap, there is some evidence that there are differences between these two stigma types [29]. To our knowledge, there is only a limited number of evaluated self-help online programs which aim at reducing suicide stigma [30, 31] or improving suicide literacy [32]. Furthermore, the involvement of persons with a lived experience in suicide prevention interventions is scarce [33]. Therefore, we developed the German online suicide prevention program *8 Lives–Lived experience reports and facts on suicide* involving persons with a lived experience of suicide [34]. The aim of this article is to present the evaluation of this newly developed program by answering the following research questions:

1. To what extent do suicide literacy, self-stigma and perceived suicide stigma, and self-efficacy expectation of being able to seek support in psychologically difficult situations of the participants change after completing the program compared to baseline?

2. How do the participants evaluate the program and aspects of the program (e.g., overall satisfaction and helpful program elements) at completion and 12 to 26 weeks after program completion?

3. What kind of changes, including adverse events or undesired side effects, do participants report 12 to 26 weeks after program completion?

## Methods

We used a mixed methods sequential explanatory design [35] to evaluate the online program. As described in our study protocol [27], the quantitative part included an online pre-post-survey with participants of the online suicide prevention program to assess possible changes over the course of the program in terms of the main goals and satisfaction with the program. The qualitative study comprised of follow-up telephone interviews with a subsample of online program completers to understand possible changes and participants' evaluation of the program in more depth.

### Ethics approval and consent to participate

The Ethics Committee of the Hamburg Medical Chamber approved this study on March 9, 2018 (process number: PV5750). All participants were informed in written form about the voluntariness of their participation, about data protection and about the possibility to terminate their participation in the online program or the telephone interview at any time. Participants were informed of the goals of the program, that it was not a crisis intervention program, and that the program was not a substitute for on-site personal care. Referrals were made to regional and telephone support services. Participants in the online program provided consent by checking an online tick box. Participants in the telephone interviews provided written informed consent to participate. All participants provided informed consent for publication of the results in anonymous form.

### Content, development process, and rationale

The content and development process of the unguided online program are described elsewhere [34]. The program was intended to be a low-threshold program that could be used anonymously by participants. An outline of the content of the program's eight chapters can be found in S1 Table. The program development closely involved ten persons with lived experiences of suicide ("lived experience team"). The program is based on the Australian digital intervention *The Ripple Effect* [30, 36]. A distinctive feature in suicide prevention is the consideration that suicide stigma, among its many negative effects, such as reduced help-seeking and self-deprecation, may be a protective factor preventing some individuals from taking their lives [37]. We therefore paid special attention not to normalize suicidality or suicides within the developed program. We also implemented a permanent access to information about support services. In this regard we also refer to the so called "Papageno effect" that appropriate media coverage of suicides can prevent them, by refraining from monocausal explanations and detailed descriptions of the circumstances and instead pointing out constructive ways of coping with crisis situations and professional support options [38], especially when the message is delivered by a person with personal experience [39]. In the program development, we refer to the empowerment and recovery story telling approaches by using video reports on lived experience of suicide. We also resorted to elements from cognitive behavioral therapy, that participants could optionally use. However, our program does not have a therapeutic focus. We classify it in suicide prevention through education and awareness.

### Eligibility criteria for participants

Criteria for participating in the online program were:
 (1) at least 18 years of age,
 (2) internet access and the ability to understand German language,
 (3a) affected by suicidal ideation or a suicide attempt, or

(3b) affected as a close person, i.e., loss of a person by suicide or caring for a suicidal person, or

(3c) interested in the topic of suicidality in general.

Participants were assigned to one of five program variants, depending on the self-reported type of affectedness that currently affects them the most. The content, e.g., texts, videos of persons with a lived experience of suicide and work sheets varied accordingly. Age was the only fixed exclusion criterion. Language skills were not assessed. As the program was neither a therapeutic nor a crisis intervention program, the current level of suicidality of the participants was not assessed.

## Recruitment

The program was available free of charge on the subdomain https://8leben.psychenet.de/ from December 19, 2019, to August 31, 2020. Study participants were recruited by various means, including teasers on the e-mental-health portal *psychenet.de*, e-mail appeals to multipliers (e.g., distribution lists of university clinics, self-help organizations) to spread the link to the study website, and references to the study in social media. Search engine optimization was performed. In addition to the online recruitment, posters, postcards, and notices were distributed in supermarkets, medical practices, and psychiatric facilities in two German cities (Hamburg, Berlin). A press release of the University Medical Center Hamburg-Eppendorf provided information on the study.

## Delivery method and setting

Participants used their own devices to access the browser-based online program (e.g., smartphones, tablets). Chapters were unlocked successively. Chapters 1 and 2 contained the baseline assessment, chapter 8 contained the post assessment (see also S1 Table). When a participant completed all eight chapters, the participant could access all chapters again as well as the library, which contained all video reports, worksheets, and a gallery of "digital postcard messages" from other participants.

## Exposure quantity, duration, and time span

We informed the participants that the program would take about 1.5 to 5 hours, and participants could freely divide the time or logout occasionally and continue working at the chapter that was last saved. There was a note to take a break at the end of chapters 3–6. Web analytics (e.g., time spent on the eight chapters) were carried out using the open-source web analytics tool Matomo; see S2 Table for results.

## Activities to increase adherence

The use of financial incentives in health research is widely debated [40]. Given the sensitivity of the topic of suicidality and suicide, we decided that it was more appropriate not to use reminder emails, financial incentives, or other compensations to encourage individuals to participate in our study, or to encourage participants to continue using the program if they dropped out.

## Outcomes

**Quantitative approachPre-post comparison without control group.** The primary outcomes were suicide literacy and perceived suicide stigma, assessed online at baseline and post-intervention using the following scales:

**Literacy of Suicide Sale–Short Form (LOSS-SF).**   We used the translated German version of LOSS-SF [24, 41] to assess suicide literacy. The scale includes twelve true and false statements concerning suicidality covering the domains signs, risk factors, causes/nature and treatment/prevention. Correctly answered statements were coded with a score of 1; false or "I don't know" responses were coded with 0. By summing the item scores, a total score can be calculated (0–12).

**Stigma of Suicide Scale–Short Form (SOSS-SF) adapted: Perceived suicide stigma.**   We used a translated and adapted German version of SOSS-SF [42, 43] to assess perceived suicide stigma. Perceived suicide stigma is defined as a person's believe about the general public's attitude (stereotypes) towards a person who dies by suicide. Consistent to the evaluation of *The Ripple Effect* [30], we used the introductory statement "In general, other people think that a person who takes his or her own life is . . .". The sixteen following descriptions were used as in the original: eight items assessing stigma (e.g. "irresponsible"), four items each assessing isolation/depression (e.g. "lonely") and normalization/glorification (e.g. "brave"). Participants agreement was measured on a 5-point Likert scale (strongly disagree to strongly agree). In the principal component analysis with varimax rotation, based on an eigenvalue greater than 1, there was a three-component solution that explained 57.7% of the variance. Scores were calculated for these three subscales. The internal consistency (Cronbach's alpha) in our sample at baseline (N = 802) was $\alpha$ = .88 for subscale stigma, isolation/depression: $\alpha$ = .82; normalization/glorification: $\alpha$ = .67.

The secondary outcomes were self-stigma and self-efficacy expectations of being able to seek support measured pre- and post-completion of the online program:

**Stigma of Suicide Scale–Short Form (SOSS-SF) adapted: Self-stigma.**   To assess negative attitudes towards oneself because of own suicidality as a part of self-stigma, we used the same 16 descriptors as in SOSS-SF but we changed the introductory statement to "Because I had thoughts of taking my life, I feel . . ." [30]. Response categories were the same as in the perceived stigma scale. The principal component analysis with varimax rotation yielded a three-component solution that resolved 55.5% of the variance, with the item "vengeful" loading substantially on the factor normalization/glorification (.38) in addition to the factor stigma (.25), yet we assigned it to the subscale stigma. The internal consistency in our sample at baseline (N = 507) was Cronbach's $\alpha$ = .81 for the subscale self-stigma, $\alpha$ = .84 for isolation/depression, $\alpha$ = .78 for normalization/glorification. Different to our study protocol [27], we considered self-stigma as a secondary outcome, as only participants from program variants 1 (suicidal ideation) and 2 (suicide attempt) answered this instrument.

**Self-efficacy expectations of being able to seek support in psychologically difficult situations (SWEP-6 and SWEP-7).**   A scale to measure self-efficacy expectations of being able to seek support in psychologically difficult situations–based on Bandura [25, 44]–was newly developed for this study [26]. With 6–7 German items on an interval scale from 0 ("I do not feel confident at all") to 10 ("I feel completely confident"), the participants were asked to indicate the extent to which they feel confident to seek support, e.g. "I can seek professional support (e.g., physician, psychotherapist) when I need it.". The additional SWEP-7 item "I feel confident that I can talk to someone about my suicidal thoughts." was only administered in program variant 1 and 2. Cronbach's $\alpha$ of the SWEP-6 scale in our sample (N = 802) was $\alpha$ = .89.

**Distress thermometer.**   As an additional measure we used the distress thermometer [45] in the pre- and post-assessment to capture current distress. Using a thermometer scaled from 0–10, participants were asked to indicate how much distress they felt during the past week, including today (0 = no distress, 10 = extreme distress).

**Satisfaction with the program and helpful elements.**   As an additional measure we assessed satisfaction with the program after completion using a 5-point Likert scale (strongly

disagree to strongly agree or not at all helpful to very helpful). In total, participants answered 21 items in the domains: a) knowledge about suicidality and stigmatization (e.g. "By participating in the online program, I know the risk factors and protective factors of suicidality better."); b) skills (e.g. "By participating in the online program, I can talk better to others about my lived experience of suicide."); c) helpful elements of the program (e.g. "Please rate how helpful you found these elements of *8 lives*: Lived experience video reports"); d) satisfaction with the length of the program; e) sharing own experiences ("This was the first time I shared my attitudes about suicide and/or lived experiences of suicide. Yes/No"); f) recommendation of the program ("Would you recommend the online program to others? Yes/No.").

## Qualitative approach: Follow-up telephone interviews

At the end of the post-assessment, study participants who completed the online program could optionally leave their contact details in an online input mask. These interested persons were contacted by e-mail approximately 6–12 weeks after completion of the online program. They received further written study information about the telephone interview and were requested to sign an informed consent to participate in the interview. 12 to 26 weeks after completing the online program, we conducted semi-structured follow-up telephone interviews, i.e., open evaluation interviews, with this subsample of completers to explore a) reasons for participating in the program, b) experiences with using the program, c) changes after participation in the program, d) evaluation of the program, e) appropriate consideration of suicide stigma experiences in the program. We developed a semi-structured guide following a process that involved (1) collecting questions, (2) reviewing the questions from aspects of prior knowledge and openness, (3) sorting the remaining questions and keywords, and (4) subsuming and bundling the questions, including finding a narrative prompt that is as simple as possible [46]. The guide can be found in S7 Table. The telephone interviews (one-on-one) were conducted by MD. Interviews were digitally audio recorded and then transcribed for further analysis according to predefined transcription rules [47]. Outcomes are the developed coding tree (see S8 Table) and participants' quotes. In S4 Table, we present further information based on COREQ checklist [48].

## Sample size for quantitative and qualitative approach

A conservative power calculation was carried out following the study on the digital intervention *The Ripple Effect* [27, 36]. As described in our study protocol, we planned a pre-post-design with no control; a sample size of 241 completers was necessary to identify an effect size of $d = .20$ with a power of .80 and a significance level of $\alpha = .05$. In the a priori power analysis, we adjusted alpha for two primary endpoints (SOSS-SF and LOSS-SF) [27]. Due to the exploratory design of the study and our aim to provide an online program that is accessible and available for all interested parties, a randomized controlled design was not conducted. Primary and secondary outcomes were tested two-sided.

In terms of the qualitative part, we planned to conduct at least twelve interviews [49]. We used maximum variation sampling with respect to different program variants, genders, and age groups to collect data from as many different perspectives as possible [50].

## Quantitative approach: Statistical methods

We calculated arithmetic means and standard deviations for primary and secondary outcomes at baseline for all participants. We performed Chi-square test, two-sided Fisher's exact test, and t-test for independent samples to explore differences between completers and non-completers. This included a comparison regarding age, gender, years of education, size of

residence, online program variant, baseline data in primary, secondary and additional out-comes. Our hypothesis was undirected; we tested whether there were differences at baseline. We used descriptive statistics to assess satisfaction with the online program at post-assessment. To explore pre-post-differences in distress, we performed the t-test for dependent samples.

Different from the study protocol, we did not use the t-test for dependent samples for primary and secondary outcomes. Instead, we used linear mixed models to calculate the difference in estimated marginal means (EMMs) from pre- to post-intervention for participants with complete data sets. Covariates included age, gender, education, size of residence, chosen variant of the online program (own affliction with suicidality), and distress since these factors can be associated with stigmatizing attitudes as well as suicide literacy and self-efficacy expectations [24, 51]. To check the robustness of the results, we also conducted a sensitivity analysis with the full data set by handling missing data with maximum likelihood estimation on all available data. For the four primary outcomes (LOSS-SF scale; perceived SOSS-SF subscales stigma, isolation/depression and normalization/glorification), we applied Bonferroni adjustment in the main analysis (completer only) to limit alpha error inflation due to multiple testing. Thus, differences on these outcomes were tested at an adjusted significance level of $p \leq$ .0125. For all other analyses, results with $p \leq$ .05 were considered statistically significant. Effect sizes for pre- to post-program changes (Cohen's d) were calculated by dividing the estimated marginal mean differences by the standard deviation at baseline [52].

Finally, in the main analysis, we exploratively compared subgroups on primary and secondary outcomes. We predefined the following subgroups: Variant of online program (1–5), gender, age groups (e.g., 18–29 years), education (lower/higher), size of residence (e.g., city), and distress (lower/higher). First, we identified differences between subgroups by inserting an interaction term. Relevant differences were defined based on the p value ($p < $ .05) of the interaction term. Second, we calculated the EMM pre-post difference within each subgroup. Due to the exploratory nature of the subgroup analyses, we did not adjust for alpha error inflation. For results, see S6 Table.

Participants had to answer every item to progress in the program, i.e., for quantitative data there were no missing values unless a participant dropped out. Analyses were performed with the statistics software IBM SPSS 27 and R version 4.2.1.

## Qualitative approach and research paradigm

We analyzed the transcribed telephone interviews according to structuring qualitative content analysis [47] following a realistic paradigm [53]. The analysis was performed in seven steps considering the research questions: (1) marking important text passages, (2) developing main themes, (3) coding the entire material, (4) compiling all text passages coded with the same main theme, (5) inductively determining subcodes on the material, (6) coding the complete material with differentiated coding tree, (7) further analyses. The first draft of the coding tree was developed by MD, was discussed during the research process and modified iteratively (JB, MD). Two researchers (MD, JB) coded main themes and subcodes for all transcripts independently using MAXQDA 18 and 20 (VERBI). To increase intersubjective reproducibility and comprehensibility, we presented and discussed the qualitative research twice with an interdisciplinary group of researchers in a qualitative methods seminar led by NP.

## Integration of quantitative and qualitative findings

The integration of the quantitative and qualitative data (carried out by MD) is shown in a joint display including descriptions how qualitative findings may help explain quantitative findings [54]. A convergence assessment is made to display the findings obtained from each component

[55]. There are four options: *agreement*, *partial agreement*, *silence*, or *dissonance* between the quantitative and qualitative results. Here, silence means that a result emerges only from qualitative or only from quantitative data.

# Results

## Participant flow

The sample size (N = 268) was achieved in 8.5 months and slightly higher than the targeted sample size (N = 241). The dropout between baseline and post-assessment was 66.6% (Fig 1). Program variant 3 (loss by suicide) showed the highest dropout (73.1%), program variant 5 (interested in the topic) the lowest (60.2%).

## Recruitment setting

The study completers stated that their attention was drawn to the study via the e-mental-health portal *psychenet.de* (n = 79, 29.5%), through search engines (n = 50, 18.7%), recommendations from friends, family members or acquaintances (n = 48, 17.9%), references on social media (n = 32, 11.9%), another website dealing with suicide (n = 28; 10.4%), references from support groups (n = 4, 1.5%), newspapers (n = 4, 1.5%), and through other means (n = 23, 8.6%).

## Periods between pre-, post- and follow-up-assessment

The time between pre- and post-assessment varied between participants depending on how long it took them to complete the online program. Between the end of the baseline survey and the end of the post-survey, participants (N = 268) took the median time of 2.5 hours (M = 102.2 hours, SD = 378.8; min: 0.3 hours, max: 3,669 hours). 70.1% of participants completed the program and pre-post-assessment within one day (Fig 2). Follow-up telephone interviews (n = 16) were conducted at a mean of 13.3 weeks (SD = 6.9; range: 3.6–31.1 weeks; median: 12.6 weeks) after post-assessment.

## Baseline data

As shown in Table 1, mainly women (n = 647, 78.2%), persons with a higher education level (n = 594, 71.8%) and persons living in a bigger city (n = 417; 50.4%) participated in the online program. Most of the participants reported having suicidal ideation (n = 337, 40.7%). There were no significant differences between participants who completed the study and non-completers regarding sociodemographic variables or distribution in the program variants. In the baseline assessment, we found significant differences between completers and non-completers in distress (non-completers showed higher distress), in two subscales of self-stigma (non-completers showed higher scores in stigma and in isolation/depression), and in one subscale of perceived stigma (non-completers showed higher scores in the normalization/glorification subscale).

## Quantitative approach: Pre-post-comparison

Table 2 shows the estimated marginal means (EMM) differences in primary and secondary outcomes in two data sets using linear mixed models: 1) subset of participants who completed the post-program survey (completer only; main analysis); and 2) full data set with missing data handled using maximum likelihood estimation (sensitivity analysis).

   **Primary and secondary outcomes.**   Participants of the online program showed a moderate to high significant increase in suicide literacy in the post-assessment ($p < .001$); on average, participants answered 1.85 more items of LOSS-SF correctly compared to baseline. The highest

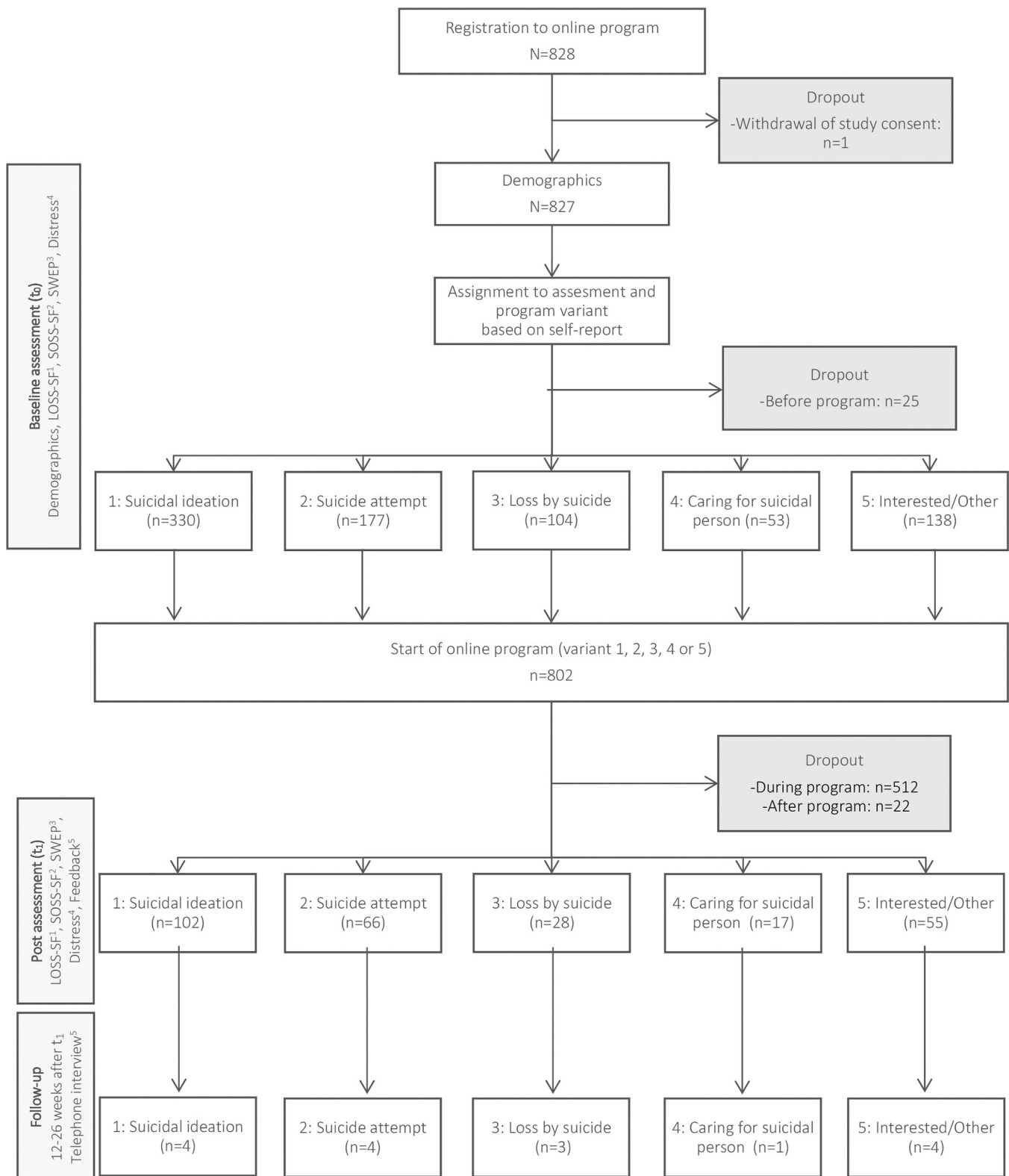

**Fig 1. Flow chart.** [1]LOSS-SF: Literacy of Suicide Scale, [2]SOSS-SF: Stigma of Suicide Scale [3]SWEP: Self-efficacy expectations of being able to seek support in psychologically difficult situations, [4]Distress thermometer, [5]Self-developed instrument.

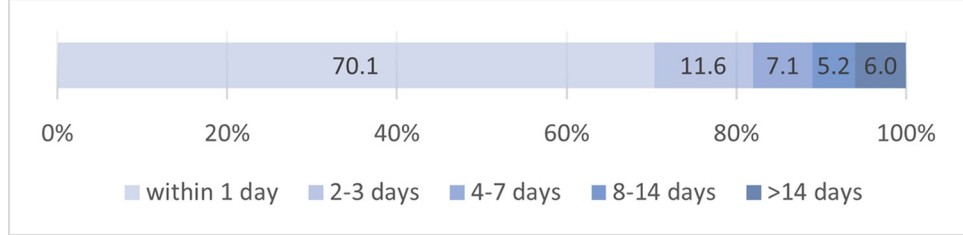

**Fig 2. Period between pre- and post-assessment by proportion of completers (N = 268).**

increase in literacy between baseline- and post-assessment was shown for the items "*A suicidal person will always be suicidal and entertain thoughts of suicide (False)*", "*There is a strong relationship between alcoholism and suicide (True)*", and "*People who talk about suicide rarely kill themselves (False)*". For perceived stigma (SOSS-SF), participants showed no significant differences between the pre- and post- assessment for all three subscales (*stigma*: $p = .62$, *isolation/ depression*: $p = .69$, *normalization/glorification*: $p = .07$).

For self-stigma (SOSS-SF), participants showed a small significant decrease compared to baseline in the subscales *isolation/depression* ($p = .04$; $d = -.14$) and *normalization/glorification* ($p = .04$; $d = -.13$). For self-efficacy expectations of being able to seek support (SWEP), participants showed a small significant increase ($p < .01$; $d = .09$) after completing the online program.

**Additional measures.** Distress between pre- and post-assessment decreases slightly by -.28 (95% CI: -.13; -.43; $t = 3.6$) in completers (pre-assessment: $M = 6.37$, $SD = 2.72$; post-assessment: $M = 6.09$, $SD = 2.75$).

**Satisfaction with the program at post-assessment.** 240 of the 268 completers (89.6%) would recommend the online program to others. Of all completers, 180 (67.2%) participants found the length of the program just right, 72 (26.9%) too long and 16 (6.0%) too short. 136 completers (50.7%) stated, that within the online program, they shared their attitudes about suicide and/or lived experiences of suicide for the first time. Participants wrote a total of 212 digital postcard messages, i.e., personal narratives on suicidality and stigma.

Fig 3 shows participants' feedback on the online program at post-assessment.

## Qualitative approach: Follow-up telephone interviews

44 participants (16.4% of the completers) agreed to be contacted for an additional telephone interview for further evaluation of the program. To reach a maximum variation sample of at least 12 participants of the 44 potential interviewees, we contacted the first 30 by mail. Of those contacted, 13 (43.3%) did not respond and one person (3.3%) cancelled because she no longer had interest in an interview. We conducted interviews with 16 participants between June and September 2020, representing a response rate of 53.3% among those contacted. As shown in Table 3, the sample shows a good variation of characteristics concerning gender and program variant. The median program completion time of the interviewed participants was 3 days ($M = 6.75$ days, $SD = 10$, range 0–40 days) and thus considerably higher than the median of all completers (Fig 2).

## Qualitative approach: Main results

Tables 4 and 5 present the results of the qualitative analysis for the subcodes of *evaluation and achievement of program objectives*. Descriptions of the corresponding subcodes and example quotes from participants illustrate the results. The entire coding tree is provided in S8 Table

**Table 1. Baseline data for total sample and compared between completers and non-completers.**

| Variable | Baseline (N = 827) | | Completers (n = 268) | | Non-completers (n 559) | | |
|---|---|---|---|---|---|---|---|
| | *n* | % | *n* | % | *n* | % | *p* |
| Gender | | | | | | | .248 [a] |
| Female | 647 | 78.2 | 202 | 75.4 | 445 | 79.6 | |
| Male | 158 | 19.1 | 56 | 20.9 | 102 | 18.2 | |
| Diverse | 22 | 2.7 | 10 | 3.7 | 12 | 2.1 | |
| Age in years, *M (SD)* | 36.47 (13.59) | | 37.2 (14.2) | | 36.11 (13.3) | | .284 [b] |
| Range | 18–79 | | 18–72 | | 18–79 | | |
| Program variant | | | | | | | .208 [a] |
| 1: Suicidal ideation | 337 | 40.7 | 102 | 38.1 | 235 | 42.0 | |
| 2: Suicide attempt | 180 | 21.8 | 66 | 24.6 | 114 | 20.4 | |
| 3: Loss by suicide | 107 | 12.9 | 28 | 10.4 | 79 | 14.1 | |
| 4: Caring for a close suicidal person | 56 | 6.8 | 17 | 6.3 | 39 | 7.0 | |
| 5: Interested/Other | 147 | 17.8 | 55 | 20.5 | 92 | 16.5 | |
| Education | | | | | | | .284 [c] |
| In school | 22 | 2.7 | 7 | 2.6 | 15 | 2.7 | |
| No school-leaving qualification | 3 | 0.4 | 1 | 0.4 | 2 | 0.4 | |
| Special-needs school | 1 | 0.1 | 0 | 0 | 1 | 0.2 | |
| Basic school qualification (9 years of high school) | 32 | 3.9 | 9 | 3.4 | 23 | 4.1 | |
| Intermediate school qualification (10 years of high school) | 163 | 19.7 | 56 | 20.9 | 107 | 19.1 | |
| Higher education entrance qualification (12–13 years of high school) | 289 | 34.9 | 104 | 38.8 | 185 | 33.1 | |
| University degree | 305 | 36.9 | 85 | 31.7 | 220 | 39.4 | |
| Other | 11 | 1.3 | 6 | 2.2 | 5 | 0.9 | |
| Not specified | 1 | 0.1 | 0 | 0 | 1 | 0.2 | |
| Country | | | | | | | .140 [a] |
| Germany | 774 | 93.6 | 248 | 92.5 | 526 | 94.1 | |
| Austria | 30 | 3.6 | 15 | 5.6 | 15 | 2.7 | |
| Switzerland | 10 | 1.2 | 1 | 0.4 | 9 | 1.6 | |
| Other | 5 | 0.6 | 2 | 0.7 | 3 | 0.5 | |
| Not specified | 8 | 1.0 | 2 | 0.7 | 6 | 1.1 | |
| Size of residence | | | | | | | .519 [a] |
| City (> 100,000 inhabitants) | 417 | 50.4 | 143 | 53.4 | 274 | 49.0 | |
| Medium sized town (20,000–100,000 inhabitants) | 152 | 18.4 | 46 | 17.2 | 106 | 19.0 | |
| Small town (5,000–20,000 inhabitants) | 123 | 14.9 | 34 | 12.7 | 89 | 15.9 | |
| Rural community (< 5,000 inhabitants) | 99 | 12.0 | 32 | 11.9 | 67 | 12.0 | |
| Not specified | 36 | 4.4 | 13 | 4.9 | 23 | 4.1 | |
| | *M* | *SD* | *M* | *SD* | *M* | *SD* | *p* [b] |
| Suicide literacy (LOSS-SF, scale 0–12) | 7.76 [1] | 2.50 | 7.68 | 2.48 | 7.80 [2] | 2.50 | .526 |
| Perceived suicide stigma (adapted SOSS-SF, scale 1–5) | | | | | | | |
| Stigma | 2.91 [1] | 0.92 | 2.87 | 0.94 | 2.92 [2] | 0.90 | .429 |
| Isolation/Depression | 3.96 [1] | 0.80 | 3.86 | 0.86 | 4.02 [2] | 0.76 | **.008** |
| Normalization/Glorification | 2.26 [1] | 0.73 | 2.22 | 0.69 | 2.28 [2] | 0.75 | .253 |

(*Continued*)

**Table 1.** (Continued)

| Variable | Baseline (N = 827) | | Completers (n = 268) | | Non-completers (n 559) | | |
|---|---|---|---|---|---|---|---|
| | *n* | % | *n* | % | *n* | % | *p* |
| Self-stigma (adapted SOSS-SF, scale 1–5) | | | | | | | |
| Stigma | 2.48 [3] | 0.85 | 2.36 [4] | 0.85 | 2.53 [5] | 0.85 | **.033** |
| Isolation/Depression | 4.13 [3] | 0.85 | 3.94 [4] | 0.96 | 4.22 [5] | 0.78 | **.001** |
| Normalization/Glorification | 2.06 [3] | 0.87 | 2.03 [4] | 0.86 | 2.08 [5] | 0.87 | .499 |
| Self-efficacy expectations (SWEP, scale 0–10) | | | | | | | |
| SWEP-6 | 6.21 [1] | 2.43 | 6.19 | 2.39 | 6.22 [2] | 2.45 | .859 |
| SWEP-7 | 5.43 [3] | 2.38 | 5.38 [4] | 2.34 | 5.45 [5] | 2.41 | .743 |
| Distress Thermometer (scale 0–10) | 6.78 [1] | 2.61 | 6.37 | 2.72 | 6.98 [2] | 2.53 | **.002** |

Higher values indicate higher suicide literacy, higher stigma, higher self-efficacy expectations, and higher distress.

[a] $\chi^2$-test.

[b] *t*-test for independent samples.

[c] Fisher's exact test due to expected cell counts of less than 5. Two-tailed tested.

[1] N = 802.

[2] N = 53

[3] N = 507.

[4] N = 168

[5] N = 339.

Significant differences based on *p* < .05 are presented in bold (not adjusted for multiple comparisons).

including all identified codes (*motivations for participating in the program*, *access to the program*, *prior knowledge or experience regarding suicide or suicidality and stigma*, *experiences during program use*, *feedback and ideas for improvement*, and *potential mechanisms of action*).

Interviewees reported an increase in their suicide literacy, and only subtle changes in suicide stigma, and self-efficacy expectations of being able to seek support in psychologically difficult situations (see Table 4). Participants also reported other changes after program participation, e.g., changes in actual action or intended action.

Overall, interview participants expressed satisfaction with the program and positively evaluated it (see Table 5). As a particularly helpful element, lived experiences video reports were highlighted.

No adverse events during or after completing the online program were reported in the telephone interviews. However, some interviewees reported that the online program was sometimes exhausting, especially some video reports were experienced as emotionally distressing. One participant reported a deterioration in mood for several days, although she did not attribute this causally to program exposure.

Three of the sixteen interviewed participants having suicidal ideation, stated that they sought help (clinic, outpatient clinic, counselling) after program completion and two associated this clearly to the program. Two interviewed participants having survived a suicide attempt stated that the program supported them to deal with and to understand the suicide attempt better.

**Table 2. Pre-post comparison in primary and secondary outcomes.**

| | | Pre-Post Mixed Models: <u>Completer Only</u> (N = 268) | | | | | Pre-Post Mixed Models: <u>Sensitivity Analysis Full dataset</u> with missing values (N = 802) | | | | |
|---|---|---|---|---|---|---|---|---|---|---|---|
| | | Pre program EMM (SE) | Post program EMM (SE) | EMM Difference (95% CI) | p | ES | Pre program EMM (SE) | Post program EMM (SE) | EMM Difference (95% CI) | p | ES |
| **Primary outcomes** | **Suicide literacy** | | | | | | | | | | |
| | *LOSS-SF* [1] | 6.79 (.32) | 8.64 (.32) | 1.85 (1.60; 2.10) | **< .001** | **.74** | 6.87 (.34) | 8.69 (.40) | 1.82 (1.60; 2.10) | **< .001** | **.73** |
| | **Perceived suicide stigma**[2] Subscales of SOSS-SF (adapted) | | | | | | | | | | |
| | *Stigma* | 2.72 (.14) | 2.74 (.14) | .02 (-.07; .12) | .62 | .02 | 2.87 (.15) | 2.90 (.15) | .02 (-.06; .11) | .59 | .03 |
| | *Isolation/Depression* | 3.80 (.13) | 3.78 (.13) | -.02 (-.11; .07) | .69 | -.02 | 3.81 (.14) | 3.76 (.14) | -.05 (-.13; .03) | .25 | -.06 |
| | *Normalization/ Glorification* | 2.37 (.11) | 2.29 (.11) | -.08 (-.16; .01) | .07 | -.11 | 2.34 (.12) | 2.25 (.13) | -.09 (-.17; -.02) | **.01** | **-.13** |
| **Secondary outcomes** | **Self-stigma**\*[2] Subscales of SOSS-SF (adapted) | | | | | | | | | | |
| | *Stigma** | 2.39 (.13) | 2.30 (.13) | -.09 (-.12; .01) | .07 | -.10 | 2.32 (.32) | 2.21 (.32) | -.11 (-.20; -.02) | **.02** | **-.13** |
| | *Isolation/Depression** | 3.91 (.15) | 3.77 (.15) | -.14 (-.26; -.01) | **.04** | **-.14** | 3.95 (.33) | 3.75 (.33) | -.20 (-.31; -.08) | **< .001** | **-.23** |
| | *Normalization/ Glorification** | 2.20 (.14) | 2.09 (.14) | -.11 (-.22; -.01) | **.04** | **-.13** | 2.19 (.33) | 2.06 (.33) | -.12 (-.22; -.02) | **.02** | **-.14** |
| | **Self-efficacy expectations**[3] | | | | | | | | | | |
| | *SWEP-6* | 6.18 (.33) | 6.41 (.33) | .23 (.09; .37) | **< .01** | **.09** | 6.83 (.37) | 7.04 (.37) | .21 (.08; .35) | **< .01** | **.09** |
| | *SWEP-7** | 6.19 (.33) | 6.44 (.33) | .25 (.06; .44) | **.01** | **.10** | 6.81 (.37) | 7.05 (.37) | .23 (.04; .41) | **.02** | **.10** |

*n = 168 for completer only; <u>n</u> = 507 for full data set. ES = Effect size Cohen's d. Significant differences (primary outcomes: *p* < .0125; secondary outcomes and sensitivity analysis: *p* < .05) are presented in bold. Covariates included age, gender, education, size of residence, variant of the online program, and distress. For each primary outcome in the completer only subset, *p*-values were adjusted for four comparisons using the Bonferroni method. *p*-values were not adjusted in the exploratory sensitivity analysis or for secondary outcomes.

[1] = scale 0–12, higher values indicate higher suicide literacy

[2] = scale 1–5; higher values indicate higher stigma

[3] = scale 0–10, higher values indicate higher self-efficacy expectations of being able to seek support in psychologically difficult situations.

## Integration

Table 6, a joint display, shows that the quantitative and qualitative results are mainly in agreement.

## Discussion

This study provided initial evidence that an online suicide prevention program can enhance suicide literacy among participants, reduce self-stigma, and promote self-efficacy expectations of being able to seek support in psychologically difficult situations. However, only minor changes were observed in the latter two. We found no significant differences between pre- and post-assessment in perceived suicide stigma. Participants reported in follow-up telephone interviews, that video reports of persons with lived experience of suicide are particularly memorable. Interviewees emphasized the importance of openness and authenticity in the video reports as well as the presentation of how to talk about suicidality and conveying hope. The program prepared participants for communication about suicidality. The encouragement for

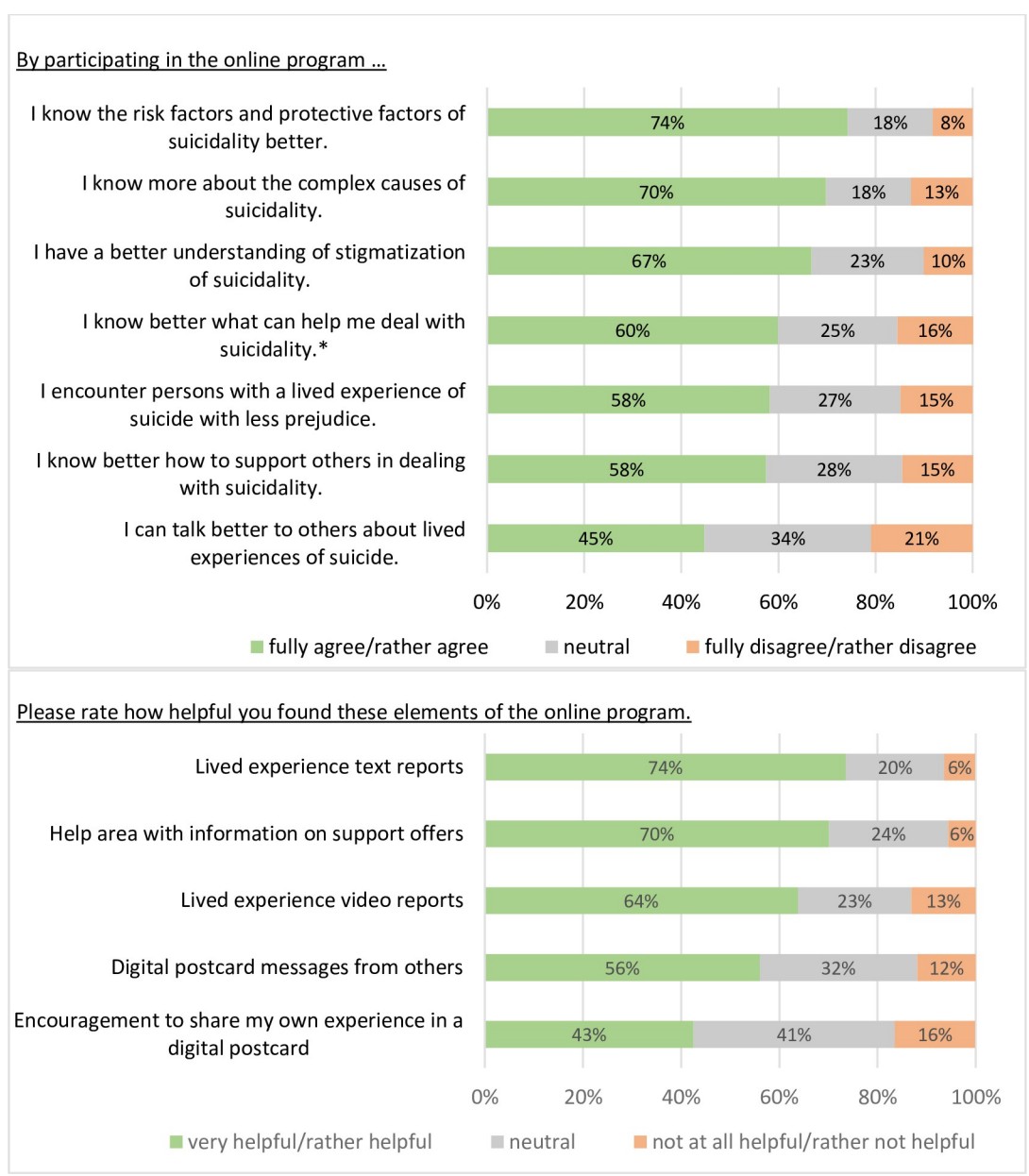

**Fig 3. Feedback and helpful elements of the online program at post-assessment (N = 268).** Self-developed items on a 5-point Likert scale at post-assessment (N = 268). The five response categories (e.g., fully agree, rather agree, neutral, rather disagree, fully disagree) are grouped into three categories in the figure. *n = 168.

self-reflection in the interactive elements, e.g., the possibility to anonymously share one's own experience of suicide, was considered as a helpful aspect as one was more actively involved. This is in line with a recent review which found that digital interventions can improve help-seeking for mental health problems especially when promoting elements of active participation by sharing one's own narrative [56]. Participants in our study reported the importance to be able to control the degree of confrontation with the topic of suicidality.

We found no substantial change in perceived stigma between pre- and post-assessment. A reason could be that the participants' beliefs about the general public attitudes is stable or at

**Table 3. Demographic data of participants, program variant, and duration of follow-up telephone interviews.**

| Variable | n = 16 | |
|---|---|---|
| | *n* | *%* |
| Gender | | |
| Female | 13 | 81.3 |
| Male | 3 | 18.8 |
| Age in years | | |
| *M (SD)* | *41.0 (10.0)* | |
| *Range* | *22–54* | |
| Variant of the online program | | |
| 1: Suicidal ideation | 4 | 25 |
| 2: Suicide attempt | 4 | 25 |
| 3: Loss by suicide | 3 | 19 |
| 4: Caring for a suicidal person | 1 | 6 |
| 5: Interested/Other | 4 | 25 |
| Education | | |
| 9 years of high school | 1 | 6.3 |
| 10 years of high school | 2 | 12.5 |
| 12–13 years of high school | 7 | 43.8 |
| University degree | 6 | 37.5 |
| Duration of interviews in minutes | | |
| *M (SD)* | *30 (12)* | |
| *Range* | *11–59* | |
| *Median* | *30* | |

least does not change because of a short intervention. We cannot rule out that some participants may even become more aware of suicide stigma in course of the program as we found a small increase in some subgroups. In principle, an increase in perceived suicide stigma by dealing with the issue is conceivable [30], so no change after an online suicide prevention program may also be a desirable effect–in particular in combination with a decrease in self-stigma. In the sensitivity analysis, we found a small reduction in the subscale normalization/glorification of perceived stigma–which is in line with our intention that through our program suicidality and suicides should neither be normalized nor glorified.

## Results compared to other studies

Evaluation results of the online program *The Ripple Effect*, on which the *8 lives* program is based, showed no significant changes in suicide literacy and stigma outcomes, except for an unexpected increase on the perceived stigma glorification/normalization subscale at post assessment [30]. In our sample, there was no significant difference in this outcome. Descriptively, we found a decrease at post assessment in the overall sample, and also in the subgroup analysis for participants affected by suicidality. The difference in evaluation results could be due to the different target samples (male farmers vs. broad target group) and to the different program content. In terms of suicide literacy, participants of *The Ripple Effect* showed higher values at baseline ($M = 9.82$, $SD = .22$) than our sample so that a ceiling effect cannot be ruled out. Although compared to a representative population survey in Germany [24], participants in our sample showed slightly higher suicide literacy at baseline ($M = 7.00$, $SD = 2.14$ vs. $M = 7.76$, $SD = 2.5$), suicide literacy still increased to post assessment, corresponding with other results from online psychoeducational programs [32].

**Table 4. Identified subcodes of main code "achievement of program objectives" and example quotes from the 16 follow-up telephone interviews 12–26 weeks after program completion.**

| Subcode | Subcode description | Example quotes |
|---|---|---|
| **Increase in suicide literacy** | Participants reported<br>• a deepening of knowledge, and a refreshing of what they have already known about suicidality and suicides, i.e., **general factual knowledge**.<br>• an increase on a more **personal and action-oriented level** (e.g., a better understanding of own suicide attempts, an increase in personal coping skills, including knowledge about help options, understanding the need to get help, and perceiving a "permission to seek support"). | *"[After participating in the program] I have much more understanding for people who try to take their live. I can understand better why they do it and I can also understand better why I tried to do it myself. So I also have more understanding of myself. (. . .) This helped me to reflect, to think and it has helped to have more compassion, compassion for others, for whatever reason they try to take their live." I13, participant who survived a suicide attempt, 12:39min* |
| **Change in suicide stigma** | Participants reported<br>• prejudices they identified through participation in the program, a reflection, and a **subtle attitude change**.<br>• an **increase in awareness of suicide stigma**, even if participants did not perceive any change in their own attitudes.<br>• the program helped somewhat in **breaking the silence** on the topic.<br>• a **relief at realizing they were not alone** in dealing with the issue of suicidality/suicide (mainly by lived experience reports and digital postcard messages).<br>Participants did <u>not</u> report<br>• an increase in self-stigma or perceived stigma. | *"[After participating in the program] it changed in a way that I don't feel like that anymore, yes, that I know that others feel the same way and that it [suicidality] is talked about or was talked about in the program. It was somehow an open way of dealing and that did me good, also the time afterwards. This 'No, we don't talk about it', I think that's really bad because then I can't get rid of my [suicidal] thoughts. [The program] was quite a relief." I14, participant who survived a suicide attempt, 09:38min*<br>*"It [participating in the program] didn't really change my attitude to the subject I would say, but rather confirmed or refreshed things." I15, participant lost a close person by suicide, 08:40min*<br>*"I don't have such a fixed point of view [regarding suicidality] and it always amazes me when people have (. . .). And so, for me, it's not a big change in attitude, but rather a greater understanding of all the different reasons why there is such a thing [suicidality]." I2, participant had general interest in the topic, 07:56min* |
| **Change in self-efficacy expectations to be able to seek support** | Participants reported<br>• **more confidence** in being able **to talk to others** about psychological problems and suicidal ideation, or in telling others how they felt. A reason could be to see others talking about suicide/suicidality in the program ('role model').<br>• already **having shown a particular behavior** in a psychologically difficult situation and therefore <u>not</u> noticing a higher self-efficacy expectation to be able to seek support in difficult situations. | *"Well, I think that [through participation in the program] I am now even more open [to seek help], that even if I was affected and somehow, (. . .) that I would tell others I don't want to live anymore or that I would have the courage to say so." I8, participant caring for a close suicidal person, 22:03 min*<br>*"Because you see in the program that people dare to talk about it, and yes, very subtly then probably, and then I also dare to open up more to this topic; I found that very good." I6, participant lost a close person by suicide, 19:24min* |
| **Other changes after program completion** | Participants reported other changes which they attributed to program participation:<br>• **Change in actual action**, e.g., talking about distress, disclosing suicidal ideation, seeking professional support, exchange more with others about the issue of suicidality and stigma in general without being affected, were more sensitive for suicide stigma when talking to others.<br>• **Change in intended action,** e.g., wanting to take more care of oneself, to implement more positive activities in everyday life, to take more responsibility for own well-being, intended to deal more sensitively or openly with people who are suicidal or if a person tells them about the suicide of a close person.<br>• **Change in the way they look at themselves or at others,** e.g., feeling more compassion for themselves, viewing their own lives as more valuable, feeling more hopeful, feeling encouraged to continue living, having a better understanding of their own suicide attempt or thoughts, feeling more compassion and having more understanding for persons who are suicidal, feeling more confident and less insecure in dealing with a suicidal person or a person who lost a close person by suicide. | *"[the program] actually encouraged me to talk about it [my problems]. So I've had phases like that for years, every now and then, and I've never talked about it with anyone before. So, neither with my family, nor with my husband. And this is the first time, and in principle it was actually through this program that I found the courage to talk about it."I12, participant with suicidal ideation, 08:41 min*<br>*"(. . .) It was also because of the program that I initiated these things [admission to a psychiatric hospital], because it was clear that action had to be taken, also for our family, yes. (. . .) [The program] has played a role in this respect, because it has set things in motion. It made it very clear that something has to happen now, because otherwise we will all slip further and yes, that I also have to change something if I want to stay alive or if I don't want to expose my children to the trauma of losing me." I5, participant with suicidal ideation, 14:10 min*<br>*"[After participating in the program] I paid more attention to my life, it was worth more to me afterwards. (. . .) [The program helped] to deal better with the fact that I attempted suicide and to deal better with my future life." I7, participant who survived a suicide attempt, 06:51min*<br>*"(. . .) there were two women [in the videos] with whom I could go along very well, one of them also conveyed quite well her perspective, the hope that was behind it. I was simply emotionally involved. It has given me courage." I5, participant with suicidal ideation, 17:27min* |

Further differentiation can be found in S8 Table (coding tree).

**Table 5. Identified subcodes of main code "overall evaluation" and example quotes from the 16 follow-up telephone interviews 12–26 weeks after program completion.**

| Subcode | Subcode description | Example quotes |
|---|---|---|
| **Positively highlighted** | Participants positively emphasized the lived experience video reports. Videos were particularly memorable. Participants reported following reasons for positive evaluation:<br>• Lived experience video reports created a **personal reference to the topic suicidality/ suicide**<br>• High **heterogeneity of persons** in video reports provided different ways of accessing the topic suicidality<br>• Perception of the videos as **honest**, partly intimate, reports, perception of **openness** in the videos as courageous<br>• Seeing that **others had managed to go on living** and what had helped them (concrete **coping options**, conveying **hope** to go on living)<br>• Seeing **how to talk about suicidality or a suicide**<br>• Noticing **not to be alone** with the topic of suicidality or loss by suicide<br>• Promoting **empathy** and made mental states of crisis more comprehensible<br>• **Length** of the interview excerpts (1–9 minutes)<br>Participants also positively emphasized<br>• **information texts on suicidality** and the overall friendly, empathetic, and praising tone in the texts,<br>• opportunities for **breaks**,<br>• general program **structure** in different chapters,<br>• possibility to anonymously **share experiences**,<br>• alternation between videos, texts, own involvement,<br>• **interactive opportunities** which initiated reflection as one was not only a recipient of information. | *"I found it quite good how the program is structured and that you progress from chapter to chapter. Especially with these breaks in between, which were quite good for me. It was also good for me to read that in principle many others feel exactly the same as I do and that there were actually people who were brave enough to have small videos made of themselves and talk about this topic [suicidality]." I12, participant with suicidal ideation, 04:54min*<br>*"In general, I think it was also attractively designed, so having videos in it is always very good in principle, if it's not just dry facts in black and white." I9, participant had general interest in the topic, 09:51min*<br>*"(. . .) and then there was another woman, oh dear (.), who keeps having suicidal thoughts. She has them again and again. And that happened to me, yes, it happens to me too and somehow, it's nice, it's not nice, but it's good when someone else has it too and you don't think you are alone, only I have something like that and I think like that and maybe I'm disturbed or something." I14, participant who survived a suicide attempt, 04:14min*<br>*"One was not only a listener, but was also actively involved in certain parts (. . .)" I2, participant had general interest in the topic, 24:09min* |
| **Negatively highlighted** | Participants reported<br>• **confrontation with the topic suicidality as demanding**, e.g., participants described it as exhausting to watch the videos (desire for more tailoring, e.g., to use an additional text option to set own pace).<br>• they were **not able to be compliant with some persons** in the video reports (e.g., a particular statement of a person or without specifying this further).<br>• **Length**: Participants found the program too long with too much information, while other participants would have liked more underpinning information. Participants preferred shorter video sequences (<2–3 minutes). | *"For me [it was] partly very, very difficult to watch these videos. I haven't watched all of them yet. For me, it would have been better to have the text, to be able to read what they're saying, what they're expressing, because when I'm reading, I can decide for myself at what pace I'm going to proceed, or whether I'm going to end after a sentence or take a break, instead of watching the videos, which was sometimes very difficult for me."*<br>*I12, participant with suicidal ideation, 04:54min*<br>*"I had the feeling that it was an additional burden to deal with the topic, so that I struggled through it for quite a while until I came to the coping strategies. (. . .) I somehow have the feeling that it's not so bad if something is burdensome if it brings me further. So that something is temporarily burdensome doesn't mean that it can't be permanently good, so I wanted to do that, burden or not." I4, participant with suicidal ideation, 06:31min* |

Further differentiation can be found in S8 Table (coding tree).

An experimental study on a video showing a person's individual recovery story after a suicide attempt showed minimal effects on treatment-seeking attitudes [57] which is in line with our qualitative findings. A study on peer-led face-to-face intervention targeting suicide-specific self-stigma of suicide attempt survivors showed promising effects [58]. This study also pointed out that a disclosure of one's suicide attempt can be associated with various risks. The online program *8 lives* dealt with advantages and disadvantages of disclosing experiences of suicidality or suicide. Therefore, it can only be compared to a limited extent to an intervention that was especially developed for disclosure. Nevertheless, our study suggests that an online program may prepare for an informed decision-making on disclosure of lived experience of suicide, and thus may encourage help-seeking. However, an online program is ultimately no substitute for a face-to-face encounter [59].

## Possible side effects

Dropout was 66.6% and higher than expected. Internet-based psychological interventions often show high dropout rates, especially unguided interventions [60]. The unguided online program presented here is not a therapeutic one, had a low threshold for participating, but at the same time was of a broad scope and comparatively complex. The lived experience video reports were described in some telephone interviews as emotionally distressing, which is consistent with prior findings [61]. In the online program *8 lives*, no concrete suicide methods nor any concrete details in this regard were mentioned. Nevertheless, engagement with the topic suicidality might have been too intense, especially for participants who reported higher levels of distress.

We decided not to exclude persons with a higher symptom burden from the program per se. We transparently informed participants that the program was not therapeutic, and not suitable in an acute crisis, about program aims, and where to find local support offers.

**Table 6. Joint display including a convergence assessment of quantitative and qualitative evaluation results of the online suicide prevention program *8 lives*.**

| | Quantitative results | Qualitative results | Convergence assessment | Integrating results |
|---|---|---|---|---|
| | **Pre-post assessment (n = 268)** | **Follow-up interviews (n = 16) Subcodes** | | **How qualitative findings can help to explain quantitative results.** |
| **Suicide literacy** | Moderate significant increase after online program in all subgroups (Cohen's d = .73) | Change in suicide literacy; Prior knowledge and prior experience regarding suicidality, suicide, and suicide stigma; Potential mechanisms of action | Agreement | *Reasons for increase*: <br> • Participants attributed the increase in suicide literacy to **lived experience video reports, information texts, and interactive elements**. <br> • Especially **interactive elements** (including evaluation questionnaires, possibilities sharing own narrative on stigma and suicidality, and personal lived experiences anonymously) stimulated **self-reflection**, which could have also led to an increase in knowledge about one's own ways of dealing with suicidality/suicide and stigma. <br> • Linking general factual knowledge with one's own experience and behavior could explain the comparatively high increase in knowledge in the quantitative pre-post-assessment. <br> • Participants reported that empathizing with persons affected by suicidality or a loss by suicide activated different emotions. This process may have led to a **better consolidation** of suicide literacy–at post-assessment and follow-up. <br> • By raising awareness of the topic suicidality and stigma, participants described having **exchanged more with others,** which could also consolidate suicide literacy. However, we cannot conclude on the actual level of suicide literacy of the participants in the telephone interviews (only participants' own evaluation of suicide literacy) at follow-up. |

*(Continued)*

**Table 6.** (Continued)

| | Quantitative results | Qualitative results | Convergence assessment | Integrating results |
|---|---|---|---|---|
| | *Pre-post assessment (n = 268)* | *Follow-up interviews (n = 16) Subcodes* | | *How qualitative findings can help to explain quantitative results.* |
| **Perceived suicide stigma** | No significant differences in perceived suicide stigma. Small, heterogeneous differences between subgroups. | Change in suicide stigma; Prior knowledge and prior experience regarding suicidality, suicide and suicide stigma; Potential mechanisms of action | Partial agreement | Overall, participants reported no, or only few, and small changes in perceived stigma in the interviews which fitted to the quantitative results. The pre-post-assessment only indirectly captured the respondent's own stigmatization tendency, as the stimulus asked about the perceived general stigmatization towards persons who take their lives. Participants did not report an increase in perceived suicide stigma, although they reported they did not know some stereotypes which were presented in the program (e.g., "cowardly"). *Reasons for no change*: • Participants described their **own attitudes towards suicidal persons as consolidated**, they became **more aware of their own attitudes and reflected on them**, but still did not notice any change in suicide stigma, which could explain quantitative pre-post-findings. • Participants reported that they had few stigmatizing attitudes anyway so that **ceiling effects** in the pre-post assessment cannot be ruled out. • One participant described she noticed own prejudices about people having suicidal ideation in terms of appearance ("someone who looks like this person in the video [attractive] can't have suicidal thoughts"). The participant reported that she was not aware of prejudices in this regard, and it was reduced by using the program, because **heterogeneous persons** report their suicidal ideation in the videos. This was one aspect which was not covered in the pre-post assessment we used and therefore could not be captured quantitatively. • One participant working with suicidal clients described that he **paid more attention to perceived stigma** and other stigma components after using the program. He did not report a change in suicide stigma itself, but **on a meta-level** more awareness for suicide stigma and a subtle change in the interaction with others. The pre-post-assessment instruments did not cover 'knowledge about different stigma components' neither 'metacognition about stigma' or 'attitudes about stigma'. |
| **Self-stigma** | Small significant reduction in self-stigma on 2 of 3 subscales (Cohen's d = -.10 to d = -.14) | Change in suicide stigma; Prior knowledge and prior experience regarding suicidality, suicide and suicide stigma; Potential mechanisms of action | Agreement | Overall, participants reported only small or no changes in self-stigma, which was consistent with the quantitative pre-post-results. Participants described a "**slight lifting of the taboo**" around suicidality. They reported the program helped to deal with self-stigma, but also with suicidality. The program was described as a "relief", also in the time afterwards. *Reasons for small reduction*: • **Encountering**, i.e., seeing persons in video reports sharing their lived experiences of suicide, • open approach to the topic of suicidality and **high degree of openness** in the videos, **authenticity of video reports** from **different persons**, • **respectful attitude** that resonates in texts, • realizing **not being alone** having suicidal ideation, made a suicide attempt, lost a close person by suicide, or caring for a suicidal person, • video reports specifically show that suicidality was talked about and how to deal with it (presentations of solution, e.g., "therapy can help"). Different constructs in the interviews intertwined, e.g., self-stigma in the context of mental disorders (such as depressive disorder or paranoid schizophrenia), and symptom burden in the context of mental disorders (e.g., extreme feelings of guilt or shame with an underlying depressive disorder). |

*(Continued)*

**Table 6.** (*Continued*)

| | Quantitative results | Qualitative results | Convergence assessment | Integrating results |
|---|---|---|---|---|
| | *Pre-post assessment (n = 268)* | *Follow-up interviews (n = 16) Subcodes* | | *How qualitative findings can help to explain quantitative results.* |
| **Self-efficacy expectations to be able to seek support in psychologically difficult situations** | Small significant increase in self-efficacy expectations (Cohen's d = .09 to d = .10) | Change in self-efficacy expectations of being able to seek support; Potential mechanisms of action | Partial agreement | Participants reported that they found the courage to talk about suicidal ideation through the program and sought support. Participants generally interested in the topic reported that the program made them more confident in dealing with the topic, with suicidal persons, and person who lost a close person by suicide. Participants stated that they did <u>not</u> notice any change in self-efficacy expectations of being able to seek support in psychologically difficult situations–because they had shown already a certain behavior. Overall, the evaluation of the interviews showed that the concept of "self-efficacy expectation" was only rarely reported without reporting concrete actions. *Reasons for increase:* • Identification possible because of **heterogeneous persons in video reports** • Persons in video reports as "**role models**" • Concrete information **texts on communication** about suicidality and on **help offers** |
| **Other changes** | N/A | Other changes | Silence | Participants reported changes after program use (see Table 4 and S8 Table) that were not captured in the quantitative pre-post assessment. |
| **Satisfaction with the program** | 89.6% of participant who completed would recommend the online program to others | Overall evaluation >Positively highlighted >negatively highlighted >neutral; Potential mechanisms of action | Agreement | Consistent with the quantitative pre-post-assessment, overall, **most participants expressed their satisfaction** with the program. *Reasons for satisfaction:* • Handing out of **coping strategies** and concrete instructions for action given • Program **conveying hope**, • Possibilities for personal **narrative sharing** and stimulation of **self-reflection** • **Reduction of insecurities** through education • **Promoting empathy** • **Involvement of persons with lived experience of suicide** in program development process • **Autonomy while using the program**, i.e., being able to control the degree of confrontation with the topic, break opportunities • For other reasons for satisfaction please see also lines above. *Reasons for dissatisfaction:* • Lived experience video reports as **too emotional distressing**; unprocessed feelings came up • Not being able to conform with all the lived experience video reports ("**some videos did not fit for me**") without specifying in detail. • The engagement with the topic and the emotional activation led to a **desire for direct exchange** (e.g., forum, chat, face-to-face), which was not possible in the program. • Some participants' **expectations were unmet**, i.e., some topics (own reasons to stay alive, support in the individual grieving process) were not sufficiently addressed in the program. • Program perceived as **too lengthy** or in the opposite **more depth** was needed. • Going though chapters several times due to **technical problems**. |

(*Continued*)

**Table 6.** (Continued)

| | Quantitative results | Qualitative results | Convergence assessment | Integrating results |
|---|---|---|---|---|
| | *Pre-post assessment (n = 268)* | *Follow-up interviews (n = 16) Subcodes* | | *How qualitative findings can help to explain quantitative results.* |
| **Adverse events, undesired side effects, dropout** | Total dropout between pre- and post-assessment: 66.6% (within intervention: 63.8%) | Adverse events; Experiences during program use; Other changes | Agreement | No participant reported **an adverse event.** Most participants reported no distress or other undesired side effects while using the online program or afterwards. Some described **dealing with the issue suicidality/suicide as unpleasant but not distressing.** *Reasons for dropouts*: • **Emotional activation**: Participants reported that during using the program they felt in parts emotionally distressed. In these cases, distress was associated mainly with the lived experience video reports but also the confrontation with suicidality in general. • Unrelated to the telephone interviews, one person who lost a close person by suicide informed us by email that he participated in the program but did not complete it. He described that receiving information and videos about persons who survived a suicide attempt was too burdening for him given his experience of losing a close person. • For other possible reasons for dropout, based on telephone interviews, see reasons for dissatisfaction (one line above) which may have led to dropouts. |

When integrating the results, it should be considered that participants in the telephone interviews engaged with the online suicide prevention program considerably longer (median 3 days, n = 16) than the average completer (median 2.5 hours, n = 268). Further explanation of subcodes can be found in the coding tree (S8 Table).

Nevertheless, we found that persons with a high distress level at baseline participated in the program. From baseline comparison and reports in the interviews, it is conceivable that these participants may have dropped out of the program, as the confrontation was too demanding, or the program was not suitable for their demands. Interview participants with higher emotional distress reported that the program did not cause them lasting distress; severe adverse events were not reported.

## Strengths and limitations

A strength of our study was its mixed methods approach combining statistical trends with personal experience of program participants [35]. By this approach, we were able to generate hypotheses about what may have contributed to the dropout and about potential mechanisms of actions. We were also able to capture changes at follow-up qualitatively, both desired and undesired, which were not determined in advance.

Our study had several limitations: Firstly, the study design without a control group did not allow any causal conclusions. Secondly, we cannot verify the data based on participants' online self-reports. Thirdly, we cannot determine whether participants watched videos or read texts. However, the web analysis showed a longer time spent in the program; given the increase in literacy at the post-assessment it seems implausible that participants did not interact with the material. Fourthly, the time period between pre- and post-assessment varied considerably. For about 70% of the participants, the post-assessment took place within 24 hours after the pre-assessment. A longer follow-up would be desirable to see if literacy effects persist. Measurement on one day has advantages, e.g., less fluctuation due to different states of conditions, but also disadvantages, e.g., program content must be processed, so that no changes are reported, or memory effects intervene. Further, from all participants who started the program, only a

third completed it, i.e., the dropout rate was high. Although we methodologically considered dropouts, the pre-post-results are only generalizable to a limited extend for those who dropped out. Other effects on the participants who dropped out cannot be ruled out based on the available data. In the qualitative evaluation, we were able to cover a broad range of statements, still we had no possibility to interview participants who dropped out of the program. For a better dropout analysis, a systematic survey of reasons for dropping out should have been conducted among all participants who dropped out, as well as a survey of their needs for the program (e.g., via e-mail and interviews). Finally, the quantitative evaluation captured perceived suicide stigma referring to persons who had taken their lives, and self-stigma of participants who had suicidal ideation or survived a suicide attempt. We did not measure other stigma types (public, anticipated, experienced stigma, stigma by association). While interpreting the quantitative results, it should also be considered that the SOSS-SF [42] asks mainly for stereotypes, while emotional and behavioral components are missing [62]. We did not assess other relevant stigma outcomes [63–66] as the scope of the questionnaire survey would have been too large, also compared to the overall length of the program.

## Implications

Reducing stigma is complicated–especially in suicide prevention–but necessary so that persons can get the support they need [14]. An online program may prepare for an informed decision-making on disclosure. Encounters with people with lived experience are essential in stigma reduction [67], but possible emotional distress for participants in the program must be considered. To address the feedback that lived experience reports can be distressing, we formulated a more detailed, yet as concise as possible, note.

For future online suicide prevention programs, we recommend a stronger tailoring as this could increase program satisfaction and may decrease emotional distress. For example, a program could consider the desired degree of confrontation (e.g., only text reports, only video reports of persons with similar kinds of experiences) in combination with current distress (e.g., adapt the sequence of modules, e.g., point out offers of help in case of high distress; introduce safety plans). Conceivable is also to consider gender, age, whether a psychiatric diagnosis is present and type of diagnosis (e.g., information and reports of persons with this diagnosis), time passed since suicidal ideation or a suicide attempt, time passed since the loss of a person by suicide, kind of relationship to person died (e.g., video reports of persons with same kind of relationship), and current level of suicide literacy (e.g., amount of basic literacy information needed). However, this would mean a larger development effort and require considerably more resources than was feasible in our pilot project.

The emotional activation in this online suicide prevention program (e.g., experiencing sadness, guilt, shame) could lead to a need for exchange with others (online or face-to-face). The development of initiatives that combine online suicide prevention programs including anti-stigma elements and the opportunity to meet a group consisting of experts by experience, their relatives, and clinicians (trialogue), or therapy groups on site would be desirable. Such an approach would combine advantages of online (high reach, low-threshold, own pace, anonymity) and offline world (actual encounter with peers, direct exchange, support offers). An online suicide prevention program planned on national level, could serve as a starting point for smaller local antistigma initiatives.

Online interventions developed for reducing suicidal ideation [28] could consider the integration of an add-on module on suicide stigma (e.g. if a person is interested in the topic and reached a sufficient stabilization) without normalizing suicidality. Such online interventions would require more supervision and guidance by a therapist.

## Conclusion

Our results suggest that an increase in suicide literacy and self-efficacy expectations of being able to seek support, as well as decreases of self-stigma can be achieved through an online program that shows video reports emphasizing hope and recovery of persons with a lived experience of suicide. However, the effects on self-stigma and self-efficacy expectations are only small. Lived experience video reports were an essential element in the online suicide prevention program as well as the possibility of interactive elements (e.g., sharing own experience on suicide). It should be noted that video reports from persons with a lived experience of suicide, but also the general preoccupation with the topic can be emotionally demanding. Therefore, we advise for future programs that participants should be able to control the intensity of confrontation within an online program, e.g., future online suicide prevention programs could consider stronger tailoring.

## Supporting information

**S1 Table. Content of the online program and variants.**
(PDF)

**S2 Table. Web analytics.**
(PDF)

**S3 Table. TREND statement checklist.**
(PDF)

**S4 Table. Consolidated criteria for reporting qualitative studies (COREQ).**
(PDF)

**S5 Table. Good Reporting of A Mixed Methods Study (GRAMMS) checklist.**
(PDF)

**S6 Table. Exploratory subgroup analyses.**
(PDF)

**S7 Table. Semi-structured guide for telephone interviews.**
(PDF)

**S8 Table. Coding tree.**
(PDF)

**S1 Data. Quantitative data pre-post.**
(XLSX)

**S2 Data. Quantitative data pre-post.**
(SAV)

## Acknowledgments

We would like to thank all participants of the online suicide prevention program. We would like to thank the lived experience team consisting of members of *Irre menschlich Hamburg e. V.*: Andre, Birgit, David, Gebke, Jenny, Ralf, Regina, among others. Finally, we would like to thank Prof. Susan Brumby and Dr. Alison Kennedy for their collaboration and the permission to use *The Ripple Effect* as a model for our program.

## Author Contributions

**Conceptualization:** Mareike Dreier, Julia Ludwig, Martin Härter, Olaf von dem Knesebeck, Johanna Baumgardt, Thomas Bock, Sarah Liebherz.

**Data curation:** Mareike Dreier, Sarah Liebherz.

**Formal analysis:** Mareike Dreier, Farhad Rezvani, Nadine Janis Pohontsch, Sarah Liebherz.

**Funding acquisition:** Martin Härter, Olaf von dem Knesebeck, Thomas Bock, Sarah Liebherz.

**Investigation:** Mareike Dreier, Johanna Baumgardt, Sarah Liebherz.

**Methodology:** Mareike Dreier, Farhad Rezvani, Johanna Baumgardt, Nadine Janis Pohontsch, Sarah Liebherz.

**Project administration:** Mareike Dreier, Sarah Liebherz.

**Resources:** Martin Härter, Olaf von dem Knesebeck, Thomas Bock, Sarah Liebherz.

**Supervision:** Martin Härter, Olaf von dem Knesebeck, Nadine Janis Pohontsch, Thomas Bock, Sarah Liebherz.

**Visualization:** Mareike Dreier.

**Writing – original draft:** Mareike Dreier.

**Writing – review & editing:** Mareike Dreier, Julia Ludwig, Martin Härter, Olaf von dem Knesebeck, Farhad Rezvani, Johanna Baumgardt, Nadine Janis Pohontsch, Thomas Bock, Sarah Liebherz.

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
