## [Decision Letter · Decision Letter 0]

22 Feb 2023

PONE-D-23-01288Evaluation of an online suicide prevention program to reduce suicide stigma: A mixed methods studyPLOS ONE

Dear Dr. Dreier,

Thank you for submitting your manuscript to PLOS ONE. After careful consideration, we feel that it has merit but does not fully meet PLOS ONE’s publication criteria as it currently stands. Therefore, we invite you to submit a revised version of the manuscript that addresses the points raised during the review process.

We look forward to receiving your revised manuscript.

Kind regards,

Mohammad Jamil Rababa

Academic Editor

PLOS ONE

Journal Requirements:

"MD, JL, NP, FR, MH, OvK, JB, TB, and SL declare that they have no competing interests."

5. We note that the original protocol that you have uploaded as a Supporting Information file contains an institutional logo. As this logo is likely copyrighted, we ask that you please remove it from this file and upload an updated version upon resubmission.

Reviewers' comments:

Reviewer's Responses to Questions

**Comments to the Author**

1. Is the manuscript technically sound, and do the data support the conclusions?

Reviewer #1: Yes

Reviewer #2: Yes

Reviewer #3: No

2. Has the statistical analysis been performed appropriately and rigorously? 

Reviewer #1: Yes

Reviewer #2: Yes

Reviewer #3: No

3. Have the authors made all data underlying the findings in their manuscript fully available?

Reviewer #1: No

Reviewer #2: Yes

Reviewer #3: Yes

4. Is the manuscript presented in an intelligible fashion and written in standard English?

Reviewer #1: Yes

Reviewer #2: Yes

Reviewer #3: Yes

5. Review Comments to the Author

Reviewer #1: Evaluation of an online suicide prevention program to reduce suicide stigma: A mixed methods study.

The introduction: The topic is directly addressed in the introduction. A clear and concise description of the research problem is provided. The main variables of the research are clearly outlined. A scientifically sound formulation of the research problem.

In regard to the goals and the importance of research, there are specific, clear, and straightforward objectives underlying the research that have been formulated in a straightforward manner. There is no doubt that research plays a significant role in the field of specialization, and it is advantageous to use it in that area.

The study was based on a number of relevant studies that were incorporated. There are a number of previous studies that have been used as the basis for this study in order to provide additional insight and information.

Research methodology considers the study's objectives. The application of systematic methods has been done in accordance with scientific standards and guidelines. In this study, the results of the research are presented in a creative, organized, and accurate way. The findings of the study are supported by sound scientific principles. It was found that the study's results are supported by those found in prior research or theories of science. The conclusion clearly reflects the research objectives and is it novel.

In the field of study, recommendations were useful. The recommendations contribute to scientific knowledge in the field of specialization. Furthermore, they were relevant to the topic and problem of the study. In the research, references were used throughout and at the end of the project to provide a sound scientific foundation for the work. Throughout the paper, references are limited to those that are cited in the text as cited in the reference list. Moreover, the writing style engages the reader and creates a sense of creativity in the reader. The research shows that the parts of the study are interconnected in a clear way.

Finally, in view of the fact that the research was conducted in accordance with the journal's standards, I recommend that this paper be accepted in its form.

Reviewer #2: Dear Authors:

Thank you for the opportunity to review the manuscript entitled “ Evaluation of an online suicide prevention program to reduce suicide stigma: A mixed methods study”. Kindly find below my comments:

- The introduction focused on the important of reducing suicide stigma and the effect of stigma on suicidality. However, the online program also evaluated suicide literacy and self-efficacy expectations in seeking support. Therefore, the introduction needs to be stretched to include these two concepts and their importance in suicide prevention. Furthermore, the title of the study needs to reflect the three outcomes (i.e., suicide stigma, self-efficacy, and literacy)

- Not sure why included people who were interested in the topic of suicidality in general? Do you think including this category affected the study results?

- How screening for participants happened? How you made sure that those who enrolled in the program met eligibility criteria?

- Does the time analytic tool used Matomo reflect that participants actually read the chapters? Did you use any kind of activity such as a pop-up quiz at to be completed towards the end of each chapter?

- You mentioned that “maximum variation sampling with respect to different program variants, genders, and age groups to collect data from as many different perspectives as possible” was used for conducting the interviews. How these sociodemographic data were collected or obtained?

- Did the participants completed the measures online or face to face? How did you know when each participant finished the program and it’s time to complete the post-survey?

- Did the baseline survey included a question on whether a participant would agree to be contacted for telephone interview?

- You mentioned from the qualitative data that some participants felt distressed watching people talking about their suicidality. What kind of preventative and supportive measure have you taken to address this issue before enrollment?

- The discussion and implications sections were well addressed.

Thank you….

Reviewer #3: The study entitled ‘Evaluation of an online suicide prevention program to reduce suicide stigma: A mixed methods study’ with the aim to develop the German online suicide prevention program 8 lives - lived experience reports and facts on suicide and to evaluate the in terms of responses, satisfaction and adverse events among the participants who undergone the program.

The manuscript could be further improved based on the following comments highlighted below.

Line 59, the eligibility criteria for participants section could be placed after the recruitment section. The exclusion criteria to be stated.

It would be good to mention whether compensation was given to the subjects for their participations.

Page 5 -7 Line 94- 127, the language version of the scales LOSS-EF, LOSS-SF, SOSS-SF, SWEP-6 and SWEP-7 to be stated e.g. German version.

Line 94, the internal consistency for LOSS-SF is to be stated.

Line 165, for the sample size calculation, more information to be stated e.g outcome variable, one or two-tailed test, comparison group etc

Line 167, more information for the interview process to be provided. e.g one-to-one interview.

Line 172, for ‘Chi-square tests, Fisher’s exact test, and t-tests’, 'tests' to be replaced with 'test'. For Fisher’s exact test, 1 or 2 tailed to be stated.

Line 207, ‘MD's coding tree’ to be rewritten.

Page 9 – 10, symbol n to be used where applicable. N to be used for overall total sample size (like in Table 1)

Table 1 footnote, 1 or 2-tailed test to be denoted.

Table 2, the post values to be presented along with pre values in a separate column apart from having the mean difference column, p-value effect size d.

Line 288-289, for T=3.6, since T is referring to Thermometer, for page 138, T for Thermometer is to be indicated. Pre distress figure to be provided.

There was duplication of Figure 1, 2 and 3.

Overall, the results were comprehensively presented. The study could have added the control group which is important for comparison. Without a control group it difficult to determine the exact effectiveness of the online suicide prevention program.

6. PLOS authors have the option to publish the peer review history of their article (what does this mean?). If published, this will include your full peer review and any attached files.

Reviewer #1: **Yes: **Adi Alsyouf

Reviewer #2: No

Reviewer #3: No

---

## [Author Response · Author response to Decision Letter 0]

22 Mar 2023

Reviewer #1

Dear Dr. Alsyouf, 

Thank you for your review! We were very delighted with your feed-back.

Reviewer #2

Dear reviewer, 

Thank you for your review and all the helpful suggestions, ideas, and hints.

-Good point! Since we defined stigma and suicide literacy as primary outcomes, we adjusted the title accordingly. As you suggested we extended the introduction (see line 87ff). Since self-efficacy expectations were predefined as secondary outcome, we did not include this concept in the title but added information in the introduction. 

-The online program was to be intended as a low-threshold program (we added this; please see line 146). The intention was to provide information through the program in the context of suicide prevention and should therefore be accessible to different participants, i.e. also persons who are not affected by suicidal ideation, a suicide attempt, or a suicide. From an antistigma perspective, it makes sense to include not only persons who are potentially stigmatized, but also those who potentially stigmatize. Furthermore, the program was not a clinical program aimed at reducing suicidality. The group of interested people were included in the evaluation as they are one target group of our program. In the subgroup analyses, we saw that there are no notable differences in terms of outcome change between the five predefined target groups (please see also Supporting Information S6 Table). There was a difference between the program variants in the amount of increase in suicide literacy; but in all program variants there was an increase in suicide literacy. Overall, the inclusion of the group of interested persons did not affect the results. 

-The online program was intended to be a low-threshold program that could be used anonymously (please see line 146). The inclusion requirements for participation were very broad, i.e. made only little restrictions (age 18 years or older, language) and solely based on self-reporting. Understanding the language is an exclusion criterion as the material was in German. However, we did not assess German language skills. Individuals who indicated an age below 18 years were referred to youth websites and could not participate further. Individuals who indicated an age of 18 or older selected the experience with suicidality that currently affects them most (line 168ff), e.g. own suicidal ideation, own suicide attempt, a close person that is suicidal or the suicide of a close person, or whether they are "just interested" in the topic without necessarily having any experience with suicidality. This was a self-assessment and was not checked by the study team. However, we think that a person is best able to assess for him/herself which experience is most important at the moment. Based on this self-assessment, a person was then assigned to one of the five program variants. 

One limitation of the study is that we cannot verify the data based on self-reporting by participants, including information on eligibility criteria. We included this point in the discussion as limitation (line 596). 

-Matomo does not recognize whether participants read the texts or watched videos. Matomo records the time when the browser called up the corresponding page. However, the program was designed in such a way that participants had to have worked through all the content chapters, i.e. "clicked through", in order to be able to carry out the post-assessment. Whether participants "only clicked through the program" or whether they actually read texts and watched videos cannot be checked - the increase in suicide literacy indicates that content has been read. We did not use any kind of activities such as pop-up quiz at the end of each chapter. Your question may have referred to the possibility of an attention control or monitoring of participants’ attention. We did not include attention control, in part to keep the program low-threshold. We included information on this in the limitation section (line 595ff). 

-In the post-assessment, participants indicated whether they would like to take part in a telephone interview and, if so, left their e-mail address. During the telephone interview, the interviewer asked for sociodemographic data. These were recorded and thus, successively, it was possible to gather insights into the program from different participants (with regard to age, gender, program variants). Please also see Supporting Information S7 table and line 282ff.

-All pre- and post-measures were collected online (the follow-up was a telephone interview). We added this information in the manuscript (line 216). The chapters of the program were successively unlocked. The post-survey was chapter 8 of the program (see also Supporting Information S1 Table) and only started when chapter 7 was completed. To make this clearer, we added accordingly in the manuscript (please see line 190).

-The question about participation in the interview was not included in the baseline assessment, but in the post-assessment (after the program was completed). This makes it impossible to contact participants who started the program but did not complete it. We added more information on the interview process; please see line 282ff.

-The qualitative data were obtained from telephone interviews at follow-up, 12-26 weeks after participants had completed the program, i.e. measurements before enrollment were not possible. However, the program was developed with persons with a lived experience of suicide and was also tested by the lived experience team in this respect. Information about local help and the national crisis hotline is available throughout the program via the "Find help" button. It was also pointed out in the study information and in various places that the program is not a crisis program and does not replace therapy or personal contact with a physician or psychotherapist. It was transparently reported in the study information and the recruitment materials what the goals of the program are and it was made clear that the program was about suicidality. Please see also line 135ff. We appreciate your time and feedback! Thank you again! 

Reviewer #3

Dear reviewer, 

Thank you for your review and all the helpful suggestions, ideas, and hints.

-Good point! The exclusion criteria derive from the inclusion criteria. Exclusion criterion was only age restriction. Language skills were not assessed. Participant’s current suicidality was not assessed as the program was neither a therapeutic program nor a crisis intervention program. We added this information in line 174ff. 

-Since the TREND Statement Checklist (please see S3 Table), which we follow in the report, lists the eligibility criteria before recruitment, we would prefer to follow this order in the manuscript. 

-Good point! No compensation was given for study participation. We discussed this issue extensively during the program development. We pointed this out in the manuscript, please see line 202ff. 

We added the German language version, please see lines 218, 225, 257. 

-The LOSS-SF (a knowledge quiz) is formed as a sum score of variables at categorial level (answer options: true/false/don't know); therefore, to our knowledge, a calculation of the internal consistency is not possible; the items are not at interval scale level. No internal consistency is reported in the corresponding paper on validation by Calear et al. (Calear AL, Batterham PJ, Trias A, Christensen H. The Literacy of Suicide Scale. Crisis. 2021.) 

-We added information on sample size calculation, please see line 300ff.

-We added more information on the interview process; please see line 282ff and line 295ff. 

-Fisher’s Exact Test was 2-tailed; we added this accordingly (please see line 314).

-“MDs coding tree” was misleading. We reworded it. Please see line 355. 

-We adjusted the symbols n and N in the manuscript. 

-We noted this in the footnote to table 1.

-We added the EMM pre- and post-values for both models. Please see table 2. 

-T was referring to t-test; we specified this in the methods section (please see line 319/320). We added the pre-assessment in distress for completers and changed T to t (please see line 446). Since there was an undirected hypothesis on changes in distress (additional measure), we exploratively examined whether there was a difference and accordingly did not report a p-value in the manuscript (p-value was p <.000).

-The program is not a clinical program; therefore, distress was not defined as a primary or secondary outcome. We would therefore prefer not show the difference between pre- and post as a figure. 

-The duplications were removed. 

-We agree with that point. Since the study is completed, there is no possibility to add a control group - in the limitations it is reported that there is no control group (please see line 595). 

We appreciate your time and feedback! Thank you again!

---

## [Decision Letter · Decision Letter 1]

12 Apr 2023

Evaluation of an online suicide prevention program to improve suicide literacy and to reduce suicide stigma: A mixed methods study

PONE-D-23-01288R1

Dear Dr. Dreier,

We’re pleased to inform you that your manuscript has been judged scientifically suitable for publication and will be formally accepted for publication once it meets all outstanding technical requirements.

Kind regards,

Mohammad Jamil Rababa

Academic Editor

PLOS ONE

Additional Editor Comments (optional):

Reviewers' comments:

Reviewer's Responses to Questions

**Comments to the Author**

1. If the authors have adequately addressed your comments raised in a previous round of review and you feel that this manuscript is now acceptable for publication, you may indicate that here to bypass the “Comments to the Author” section, enter your conflict of interest statement in the “Confidential to Editor” section, and submit your "Accept" recommendation.

Reviewer #1: All comments have been addressed

Reviewer #3: All comments have been addressed

2. Is the manuscript technically sound, and do the data support the conclusions?

Reviewer #1: Yes

Reviewer #3: Partly

3. Has the statistical analysis been performed appropriately and rigorously? 

Reviewer #1: Yes

Reviewer #3: Yes

4. Have the authors made all data underlying the findings in their manuscript fully available?

Reviewer #1: Yes

Reviewer #3: Yes

5. Is the manuscript presented in an intelligible fashion and written in standard English?

Reviewer #1: Yes

Reviewer #3: Yes

6. Review Comments to the Author

Reviewer #1: Evaluation of an online suicide prevention program to improve suicide literacy and to reduce suicide stigma: A mixed methods study

Authors in this manuscript discuss low-threshold e-health approaches for reducing suicide stigma. Using a mixed methods approach, the authors developed an online program containing video reports on the lived experience of suicide and evidence-based information on suicide. A linear mixed model was used to examine pre- and post-changes in suicide literacy, suicide stigma (self and perceived), self-efficacy and expectation of seeking support when confronted with psychologically difficult circumstances. Lastly, it should be noted that the research was conducted in accordance with the journal's standards, and the author responded well to the comments made. I recommend this paper be accepted.

Congratulation to all authors.

Best Regards

Adi Alsyouf

Reviewer #3: (No Response)

7. PLOS authors have the option to publish the peer review history of their article (what does this mean?). If published, this will include your full peer review and any attached files.

Reviewer #1: **Yes: **Adi Mohammad Ramzi Alsyouf

Reviewer #3: No

---

## [Editor Report · Acceptance letter]

20 Apr 2023

PONE-D-23-01288R1 

Evaluation of an online suicide prevention program to improve suicide literacy and to reduce suicide stigma: A mixed methods study 

Dear Dr. Dreier:

I'm pleased to inform you that your manuscript has been deemed suitable for publication in PLOS ONE. Congratulations! Your manuscript is now with our production department. 

Kind regards, 

on behalf of

Dr. Mohammad Jamil Rababa 

Academic Editor

PLOS ONE